materials science/health and disease and epidemiology

polyacrylonitrile fibre, sulfhydryl, chelation, Ag ion, antibacterial

**Authors for correspondence:**
Lining Zhao
e-mail: csbtzln@163.com
Defang Li
e-mail: 13873129468@126.com

†These authors contributed equally to this work.

This article has been edited by the Royal Society of Chemistry, including the commissioning, peer review process and editorial aspects up to the point of acceptance.

# Preparation of polyacrylonitrile-based fibres with chelated Ag ions for antibacterial applications

Li Chang[1,†], Wenjie Duan[2,3,†], Anguo Chen[1], Jianjun Li[1], Siqi Huang[1], Huijuan Tang[1], Gen Pan[1], Yong Deng[1], Lining Zhao[1], Defang Li[1] and Liang Zhao[2]

[1]Institute of Bast Fibre Crops, Chinese Academy of Agricultural Sciences, 410205 Changsha, Hunan, People's Republic of China
[2]Institute of Chemistry, Henan Academy of Sciences, 450003 Zhengzhou, Henan, People's Republic of China
[3]School of Materials Science and Engineering, Zhengzhou University, 450000 Zhengzhou, Henan, People's Republic of China

 LC, 0000-0002-9513-7637; LZ, 0000-0001-8182-4818;
DL, 0000-0002-1022-4046

The need for an excellent antibacterial material that is sufficiently powerful to never develop bacterial resistance is urgent. In this study, a series of novel polyacrylonitrile-based fibres with chelated Ag ions (referred to as Ag-SH-PANF) were prepared by a two-step chemical modification process: grafting and chelating. The properties of the as-prepared Ag-SH-PANF were characterized by Fourier transform infrared (FT-IR) spectroscopy, thermogravimetric analysis (TGA), X-ray diffraction (XRD), scanning electron microscopy (SEM) and X-ray photoelectron spectroscopy (XPS). The antibacterial activities of Ag-SH-PANF were examined against pathogenic bacteria, and an antibacterial mechanism was explicated based on the release of Ag ions from the fibres' surfaces. The results showed that, although chelation occurred between the Ag ions and the grafted amino, sulfhydryl and disulfide groups, Ag-SH-PANF retained its fine microstructure and thermal stability. Moreover, Ag-SH-PANF displayed excellent antibacterial ability against pathogenic bacteria as well as good washing durability. In terms of the antibacterial mechanism, Ag ions are the main bactericidal agents in the role of catalysts and are not consumed in the antibacterial process. Nonetheless, a relatively higher concentration of Ag ions can accelerate the bactericidal process.

# 1. Introduction

In our daily lives, we are inevitably exposed to harmful microbes that can grow and propagate rapidly under suitable environmental conditions; these pathogens can even transmit diseases through interpersonal contact and endanger our health [1]. Textile fibres are excellent habitats for these microorganisms and can propel the spread of disease because of their hollow structures [2]. Endowing fibres with antimicrobial abilities can not only kill bacteria but also protect the textile against the degradation caused by moulds [3]. There are many ways to impart antibacterial properties to fibres; among which, chemical modification has become the main focus in chemical and biomedical research. Due to the strong binding between antibacterial agents and fibres, the as-prepared antibacterial fibres maintain superb and long-lasting antibacterial effects [4]. Various antibacterial components are used for preparing antibacterial materials, such as quaternary ammonium groups [5], quaternary phosphonium groups [6,7], silver nanoparticles [8,9], silver ions [10,11], extracts from *Chloranthus henryi* [12] and capsaicin [13]. Further, silver is the oldest antibacterial agent known to humans, having been used for more than two thousand years [14]. To date, silver has been used in various forms such as silver ions, silver nanoparticles and silver complexes, or implemented in composite materials [1,15–17]. Moreover, it has been employed in many applications, including dental and medical implants, the healing of burn wounds, cosmetics, food packaging, water and air purification, and domestic appliances [18–23]. The widespread use of silver is attributable to its broad spectrum of antibacterial activity, and more importantly, its low toxicity towards mammalian cells [24]. In addition, silver has the advantages of thermal stability, low volatility, and biocompatibility, among others [8]. Furthermore, unlike antibiotics, silver does not incur obvious bacterial resistance and has good antimicrobial activities against common drug-resistant bacteria and fungi [25]. Currently, there is an urgent need to develop a novel antibacterial material with excellent activity that is powerful enough to outpace bacterial evolution.

Polyacrylonitrile fibres (PANF) are excellent flexible materials that perform very similarly to wool and, hence, are known as 'synthetic wool'. PANF have good sunlight resistance, weather resistance and mechanical strength, and they are mainly used to fabricate objects for everyday use such as wool articles, blankets, sportswear, bulky yarns, hose tubes and parasols. PANF have a tight molecular structure and good chemical stability [26,27], and they bear a large number of strongly polar cyano groups (–CN) on their macromolecular chains, which can be conveniently converted into other groups such as amides, carboxylic acids, amidoximes and hydrazides [28,29]. This makes PANF an ideal raw material to prepare functional fibres, such as amine [30], quaternary ammonium [31], quaternary phosphonium [7], sulfhydryl [32], carboxyl [33], carbon [34–37] and amidoxime [38] fibres, by chemical modification. According to the hard and soft acid-base theory, silver ions belong to the category of soft acids, which easily coordinate with soft bases. Besides, many studies have reported that a sulfur-containing group as a typical soft base can efficiently coordinate with silver, mercury and other soft acids [32,39,40]. Therefore, chemical modification is a good method to obtain stable silver containing functional materials through the coordination reaction between sulfur-containing functional groups and silver.

The aim of this study was to explore the possible antibacterial applications of modified PANF. Novel PANF with chelated Ag ions (Ag-SH-PANF) were synthesized by a two-step procedure in which amino, sulfhydryl and disulfide groups were first introduced and exposed on the surface of PANF by grafting with cysteamine, and then Ag ions were fixed to the resulting –SH and –NH moieties by chelation. Thus, Ag-SH-PANF with different Ag contents were prepared using the two-step procedure. The chemical structure and surface topography of Ag-SH-PANF were characterized by Fourier transform infrared (FT-IR) spectroscopy, thermogravimetric analysis (TGA), scanning electron microscopy (SEM) and X-ray photoelectron spectroscopy (XPS). The amount of Ag ions released from Ag-SH-PANF in different soaking solutions was also examined by atomic absorption spectrometry (AAS), and finally, the antibacterial activities of different Ag-SH-PANF against common pathogenic bacteria were tested.

# 2. Material and methods

## 2.1. Material and reagents

PANF (average fibre length = 30 mm, fibre diameter = 17 µm, acrylonitrile content ≥ 90%, and linear density = 1.92 dtex) were supplied by SINOPEC Anqing Petrochemical Co., Ltd, China. Cysteamine hydrochloride (98% purity) was purchased from Wuhan Yuanda Hongyuan Co., Ltd, China. Ethylene

**Scheme 1.** Reaction mechanism for the preparation of Ag-SH-PANF.

glycol (analytical grade) was supplied by Luoyang Tiancheng Chemical Reagent Co., Ltd, China. Silver nitrate ($AgNO_3$, analytical grade) was purchased from Beijing Chemical Works. Sodium chloride, peptone and beef extract powder were supplied by Beijing Aoboxing Biotechnology Co., Ltd, China. Pathogenic *Escherichia coli*, *Staphylococcus aureus* and *Candida albicans* were provided by the Institute of Biology, Henan Academy of Sciences, LLC, China. Deionized water was used in all experiments.

## 2.2. Preparation of polyacrylonitrile fibres with chelated Ag ions

PANF with chelated Ag ions (Ag-SH-PANF) were prepared by modifying and optimizing the reaction conditions (i.e. solvent, liquid–solid ratio and reaction temperature) reported by Duan *et al.* [41], in order that a larger scale (greater than 20 g) synthesis of Ag-SH-PANF could be reliably performed. The procedure included two steps: grafting and chelating. A certain amount (approx. 15 g) of cysteamine hydrochloride was dispersed in ethylene glycol (650 ml), and the resulting solution was adjusted with sodium hydroxide (5.3 g) to render it weakly alkaline. Subsequently, PANF (15 g; the precision scale was set to 0.0001 g) were added into the solution, and the system was reacted for 4 h at 135 ± 2°C. After the reaction, the grafted SH-PANF were washed with deionized water until the pH of the washing effluent was neutral. Finally, the samples of SH-PANF (1.0 g, the precision scale was set to 0.0001 g) were soaked in different $AgNO_3$ solutions with volumes ranging from 25 to 1000 ml and a set Ag ion concentration of 1000 mg l$^{-1}$, shaken at a constant temperature (25°C) on a mechanical oscillator for 120 min and then washed with deionized water until no more Ag ions could be detected in washwater. The obtained Ag-SH-PANF were dried to constant weight at 40°C. The reaction mechanism is shown in scheme 1. The weight gain ($\omega\%$) after each reaction step was gravimetrically determined, and calculated with the following relationship:

$$\omega\% = \frac{W_1 - W_0}{W_0} \times 100, \qquad (2.1)$$

where $W_0$ and $W_1$ are the weights of the fibres before and after modification, respectively.

## 2.3. Characterization

FT-IR spectra were collected with a Nicolet IR200 spectrometer (Thermo Electron Scientific Instruments, USA) in the range of 4000–400 cm$^{-1}$; samples were prepared by the KBr pellet method. SEM (JSM-7500F, JEOL, Japan) was performed to analyse the surface morphologies of the raw and prepared fibres after coating the samples with platinum by spray deposition for 90 s under a 5 mA current. The thermal stability of the fibres was characterized by TGA (LABSYS Evo, Setaram, France); samples were heated from 60 to 800°C at 10°C min$^{-1}$ under a constant flow of $N_2$. XPS profiles for SH-PANF and Ag-SH-PANF were recorded (Escalab 250Xi, Thermo, USA) using a monochromatic Al-K$\alpha$ X-ray source. XPSPEAK 4.1 software (Informer Technologies, USA) was used to fit the peaks to calibrate the high-resolution XPS profiles.

## 2.4. Antibacterial activity assay

Representative microorganisms (*E. coli*, *S. aureus* and *C. albicans*) were used to evaluate the antimicrobial activities of the fibres in this study. During the culture of *E. coli* and *S. aureus*, media composed of 5 g l$^{-1}$ sodium chloride, 5 g l$^{-1}$ beef extract and 10 g l$^{-1}$ peptone, at a final pH of 7.2, were used. The medium used for *C. albicans* culture was composed of 200 g l$^{-1}$ potato and 20 g l$^{-1}$ glucose. After the pathogens

**Table 1.** Ag-SH-PANF with different Ag contents.

| Ag-SH-PANF | adsorption volume (ml) | Ag (%) | adsorption capacity of SH-PANF for Ag (mg g$^{-1}$) |
|---|---|---|---|
| 1 | 25 | 2.40 | 24.59 |
| 2 | 50 | 4.68 | 49.13 |
| 3 | 100 | 8.60 | 94.09 |
| 4 | 150 | 10.93 | 122.70 |
| 5 | 200 | 12.91 | 148.24 |
| 6 | 300 | 15.11 | 178.01 |
| 7 | 400 | 15.96 | 189.87 |
| 8 | 500 | 17.13 | 206.65 |
| 9 | 600 | 17.21 | 207.91 |
| 10 | 1000 | 17.46 | 211.58 |

were revived, a volume of 1.5 ml of bacterial culture was transferred and centrifuged at 4°C for 2 min at 12 000 r.p.m. The precipitate pellet was washed with sterile saline and then evenly dispersed in sterile saline (20 ml) to produce the bacterial suspension. Ag-SH-PANF (0.1 g) were immersed in the as-prepared bacterial suspension and the system was incubated for 24 h at 180 r.p.m. and 37°C. Subsequently, bacterial suspension samples (1 ml) were taken out and diluted differently (from $10^{-5}$ to $10^{-7}$). Thereafter, the as-prepared bacterial suspensions (100 µl) were inoculated on the sterile agar plates. Finally, the plates were cultured at 30°C for 48 h (for *C. albicans*) or 37°C for 24 h (for *E. coli* and *S. aureus*) in an incubator. Meanwhile, a negative control group (no fibre in bacterial suspension) was prepared. The living colonies were counted, and each sample was analysed three times. The antibacterial ratio (%) was calculated using the following equation:

$$\text{antibacterial ratio } (\%) = \frac{A - B}{A} \times 100, \qquad (2.2)$$

where $A$ and $B$ are the living colony numbers obtained from the negative control experiment (no fibres) and the testing group, respectively.

## 2.5. Release of Ag ions from polyacrylonitrile fibres with chelated Ag ions

The concentration of Ag ions in solution after soaking the modified fibres was determined to evaluate the Ag ions released from Ag-SH-PANF. Ag-SH-PANF (0.1 g) with varying Ag contents were added into different soaking solutions (20 ml) and oscillated in a thermostatic oscillator at 180 r.p.m. and 37°C. Except for the experiments to determine the effects of different soaking periods on the release of Ag ions, the soaking period was set to be 24 h for all other experiments concerning the effects of different soaking solutions on the release of Ag ions. Physiological saline composed of 0.9% NaCl solution and *S. aureus*, *E. coli* and *C. albicans* solutions were the prepared corresponding bacterial suspensions (see §2.4) used as the soaking solutions. After each time interval, the Ag ion concentration in each solution was tested via AAS (ZEEnit 700, Analytik Jena, Germany).

# 3. Results and discussion

## 3.1. Preparation of fibres with chelated Ag ions

SH-PANF were prepared successfully by grafting cysteamine on PANF. We obtained a maximum weight gain ratio of 43.4% for SH-PANF, with a sulfur content of 3.43 At.% (figure 7a, 2.57 mmol g$^{-1}$, calculated with hydrogen ignored), as observed through XPS analysis. Ag-SH-PANF samples were also prepared successfully, promoted by the reaction of the fibres with aqueous AgNO$_3$. Different adsorption capacities and silver contents were achieved for different batches by changing the volume of the AgNO$_3$ solution supplied to the fibres; the corresponding adsorption capacities and Ag contents are shown in table 1 and figure 1. The maximum Ag adsorption capacity was 211.58 mg g$^{-1}$, and the

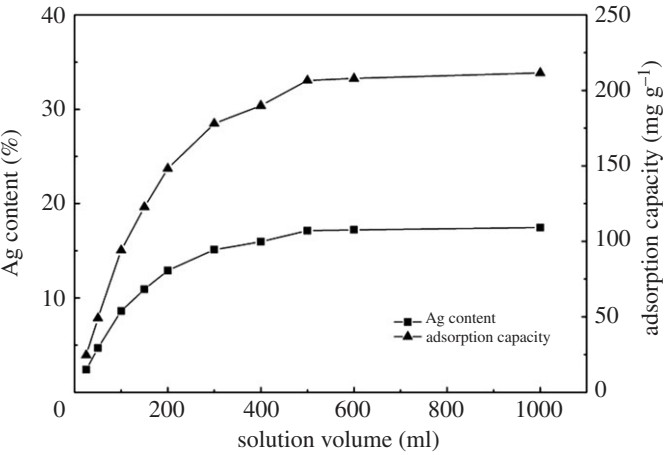

**Figure 1.** Effects of solution volume on the Ag content of Ag-SH-PANF.

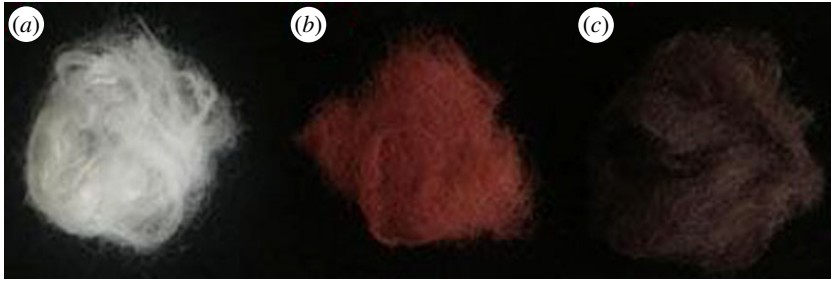

**Figure 2.** Fibre morphologies before and after modification.

corresponding Ag content of Ag-SH-PANF was calculated up to 17.46% according to the adsorption capacity. Moreover, the morphologies of the fibres before and after modification did not change significantly, although their colour shifted from the original white to a relatively dark shade, and from brownish-red to dark brown depending on Ag incorporation, as shown in figure 2.

## 3.2. Structure of the fibres

Infrared spectroscopy is a reliable method for the analysis and identification of the structure of a substance. Figure 3 shows the FT-IR spectra of PANF, SH-PANF and Ag-SH-PANF. Clearly, the spectra of the modified fibres are significantly different from that of the unmodified PANF. For SH-PANF, the absorption peak from 3150 to 3700 cm$^{-1}$ becomes relatively strong and wide, which results from the overlapping of the signal of O–H with that of N–H in cysteamine [42]. In addition, the absorption peak at 2243 cm$^{-1}$ is significantly reduced in the spectrum of SH-PANF, indicating a reduction in the number of exposed –CN groups. The absorption peak at 1732 cm$^{-1}$ disappears, and that at 1661 cm$^{-1}$ becomes relatively strong and wide, even overlapping with the peak at 1630 cm$^{-1}$, which can be ascribed to the hydrolysis of ester groups in methyl methacrylate, methyl acrylate or 2-methylene-1,4-succinic acid, the second or third monomers in PANF and the introduction of a great quantity of amino groups during the grafting process, derived from cysteamine. These results indicate that cysteamine was successfully grafted onto the fibres, and that the –CN groups were the reactive sites. For Ag-SH-PANF, the corresponding spectrum shows a narrower absorption peak at 3150–3700 cm$^{-1}$ and a stronger and sharper peak at 1384 cm$^{-1}$ that can be ascribed to the red-shift of the peak at 1389 cm$^{-1}$ (C–N) in the spectrum of SH-PANF. In addition, a new weak absorption peak appears at 2424 cm$^{-1}$ in the spectrum of Ag-SH-PANF, which belongs to the sulfhydryl group [43]; on the other hand, this peak does not appear in the spectrum of SH-PANF. Moreover, a new weak absorption peak appears at 520 cm$^{-1}$ in the spectrum of SH-PANF, which belongs to disulfide group [43]. However, this peak disappears in the spectrum of Ag-SH-PANF. All these phenomena can be attributed to the formation of complexes between amino, sulfhydryl and disulfide groups with silver. Figure 4 gives the XRD spectra of PANF, SH-PANF and Ag-SH-PANF; the fibres are completely

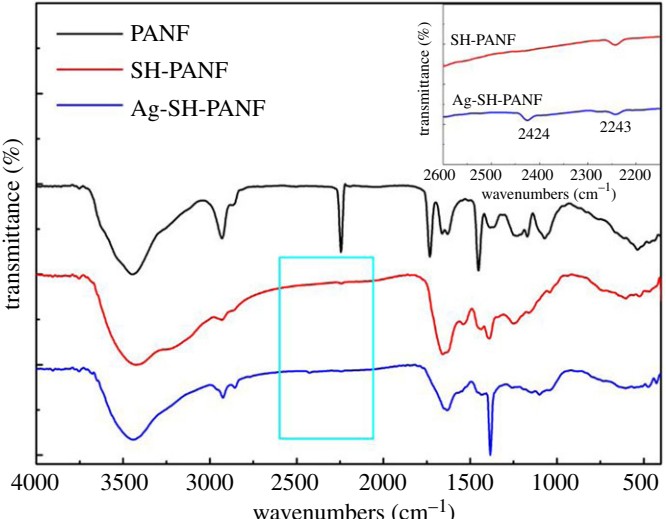

**Figure 3.** FT-IR spectra of PANF, SH-PANF and Ag-SH-PANF.

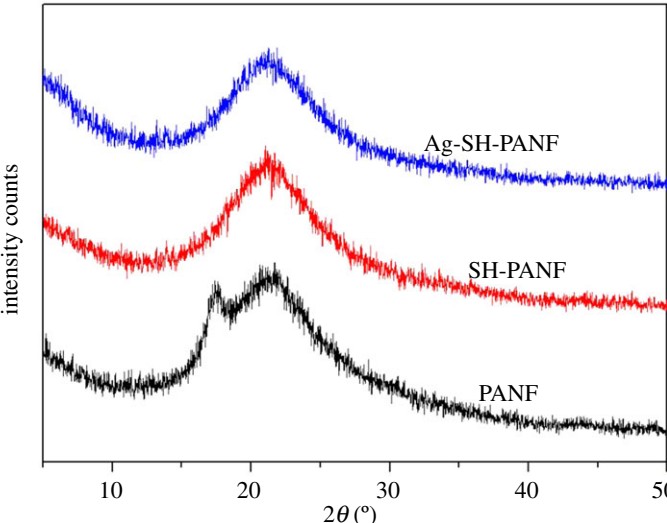

**Figure 4.** XRD spectra of fibres.

converted into amorphous structures after modification, and no obvious diffraction peak appears after silver adsorption. This shows that no silver crystals were produced on Ag-SH-PANF.

## 3.3. Thermal properties

The TGA results for PANF, SH-PANF and Ag-SH-PANF are presented in figure 5a,b. It can be easily seen from figure 5b that the weight loss rate of PANF is much higher than those of the other two fibres. According to figure 5a, in the case of PANF, there are two distinct plateaus in the TGA curve, with the first decomposition temperature near 290°C. The cyclization of –CN groups contributes to the first decomposition plateau, and the formation of a cross-linked structure in response to the oxidation and dehydrogenation of the functionalities on the macromolecular chains contributes to the second decomposition plateau [44]. After the desorption of atmospheric gases and the loss of water near 120°C, only one prominent plateau is observed in the curve for SH-PANF, whereas two plateaus appear again for Ag-SH-PANF. The second plateaus in SH-PANF and Ag-SH-PANF correspond to the first plateau of PANF. However, the decomposition temperatures for SH-PANF and Ag-SH-PANF are much lower than that of PANF, which is ascribed to the consumption of most –CN groups and the simultaneous conversion to other functionalities (i.e. –C=N) during the grafting process. The third plateau reflects the behaviour of the grafted cysteamine molecules and the cross-linked, oxidized and

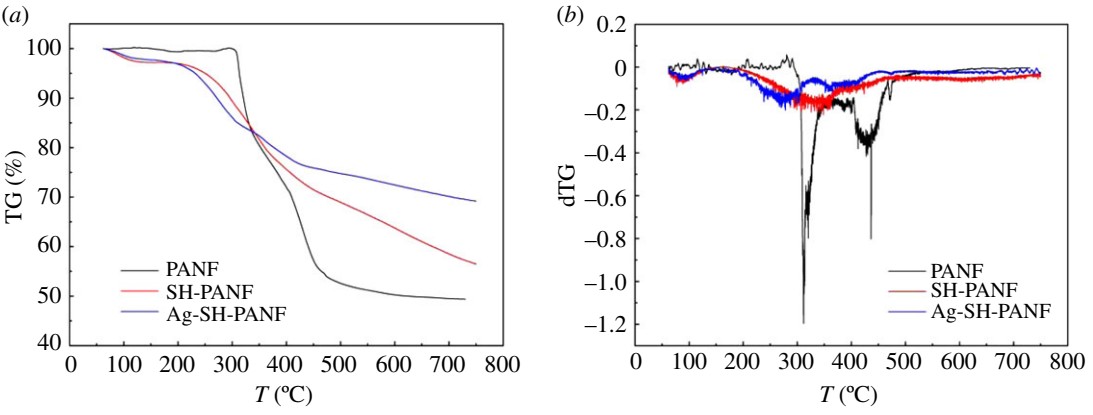

**Figure 5.** TG (*a*) and dTG (*b*) curves of PANF, SH-PANF and Ag-SH-PANF.

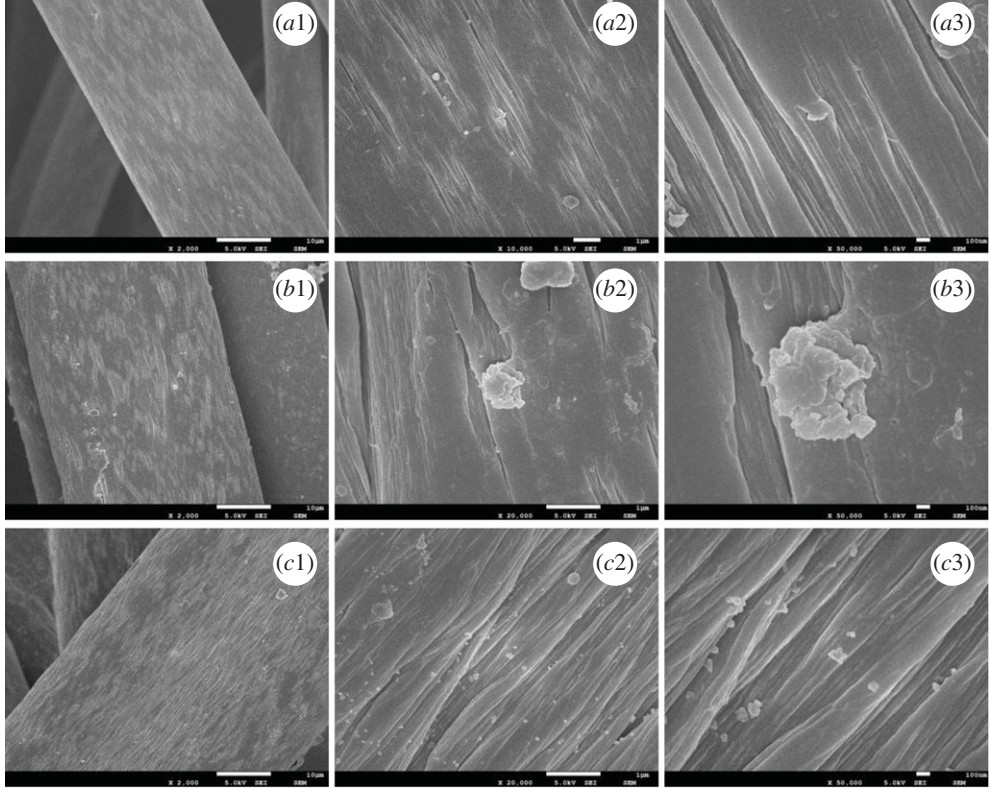

1: 2000×; 2: 20 000×; 3: 50 000×

**Figure 6.** SEM images of PANF (*a*), SH-PANF (*b*) and Ag-SH-PANF (*c*).

dehydrogenated macromolecular chains. However, after the whole process, Ag-SH-PANF gives a significantly higher char yield of 69.2% than SH-PANF (56.5%) and PANF (49.4%), which may be attributed to the large amounts of adsorbed silver. This finding accords with those of Duan *et al.* [32], but are contrary to those stated by Sun *et al.* [45], who observed that the char yield of the unmodified fibres was higher than that of heavy-metal-loaded fibres.

## 3.4. Surface morphology

As mentioned earlier, the grafting of cysteamine on PANF introduces amino, sulfhydryl and disulfide groups on the fibre surface, and the chelation of silver on SH-PANF enriches the fibres' surface with Ag ions. Figure 6 shows the SEM images of PANF (*a*), SH-PANF (*b*) and Ag-SH-PANF (*c*), which

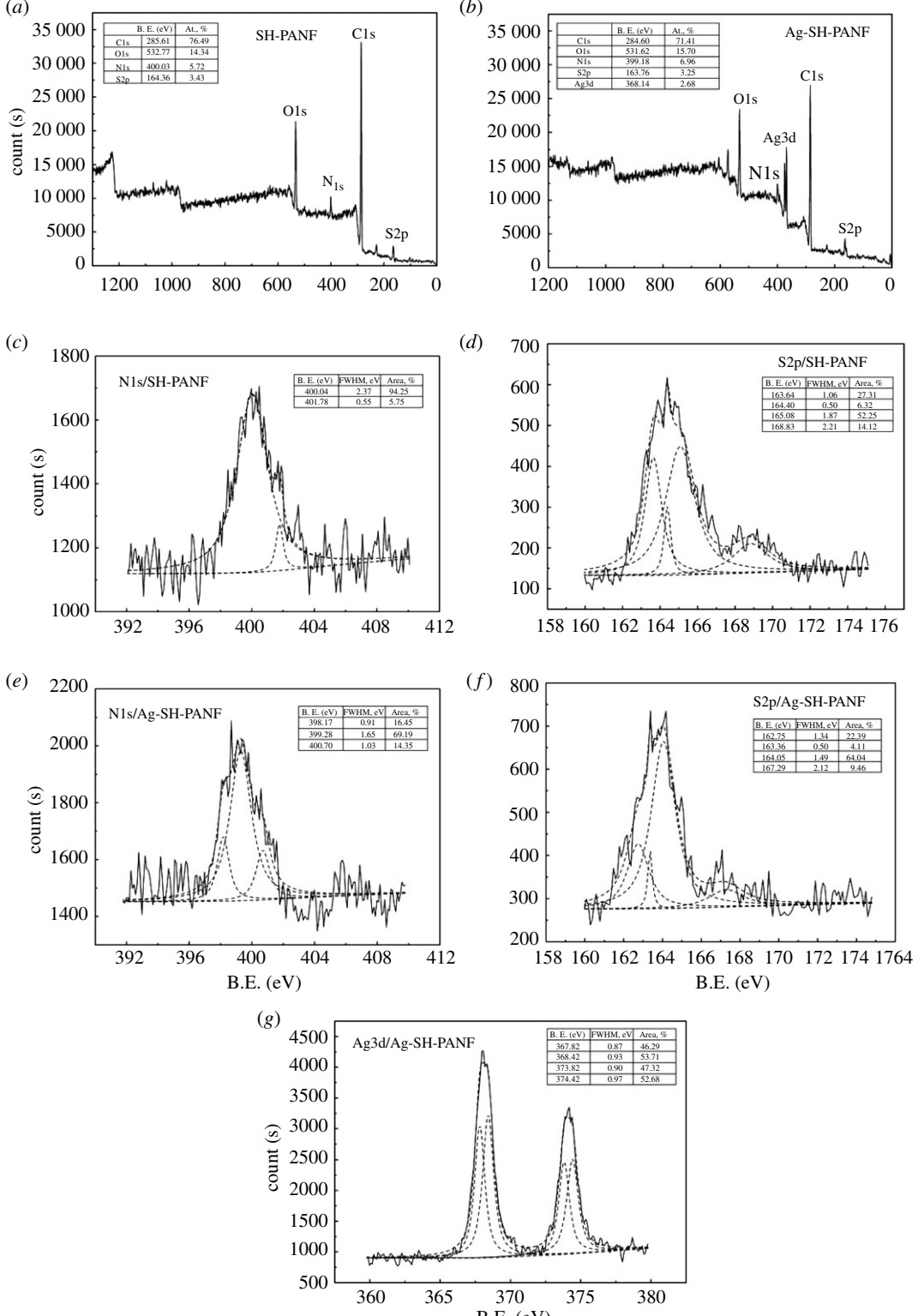

**Figure 7.** XPS survey scans and high-resolution scans of the N1s, S2p and Ag3d peaks for the fibres.

reveal that the average diameter of SH-PANF is larger than that of PANF. This is probably attributed to the swelling of the fibres during the grafting reaction [46]. The fibres' surface morphology becomes rough rather than smooth after modification, which is ascribed to the introduction of the cysteamine residues and Ag ions. Compared with unreacted PANF, more crevices are apparent on the modified fibres. Moreover, obvious randomly distributed protrusions appear, indicating that chemical

modification helps to increase the surface area of the fibres, which enhances their adsorption properties but diminishes their mechanical properties. However, since the crevices are evenly distributed on the surface of the fibres, the overall macroscopic structure is maintained after the reaction.

## 3.5. Surface binding state and elemental speciation

XPS is a very effective technique for the qualitative and quantitative analysis of elements, solid surface analysis and structural analysis of compounds. The surface binding states and elemental speciation of SH-PANF and Ag-SH-PANF were analysed by both XPS survey scans and high-resolution scans, and the data are presented in figure 7. The XPS profiles confirm the presence of S on the SH-PANF surface, while binding energies (B.E.) corresponding to Ag3$d$ were detected exclusively on the Ag-SH-PANF surface (figure 7$a$,$b$). This indicates the successful loading of Ag on SH-PANF. Deconvolution of the high-resolution XPS profile of N1$s$ for SH-PANF before and after Ag chelation (figure 7$c$,$e$) reveals that the major N1$s$ peak comprises two signals (i.e. 399.28 and 400.70 eV) that can be attributed to the nitrogen atoms of amino and cyano groups, respectively [32,46]. Deconvolution of the high-resolution XPS profile of S2$p$ (figure 7$d$,$f$) reveals the presence of multiple peaks, assignable to the sulfur atoms in sulfhydryl, disulfide and sulfo groups [47,48] at 163.64 and 165.08 eV, 164.40 eV and 168.83 eV, respectively. It should be emphasized that the two peaks at 163.64 and 165.08 eV, which shift to 162.75 and 164.05 eV after Ag chelation, belong to S2$p_{3/2}$ and S2$p_{1/2}$ in sulfhydryl groups (similar values were reported by Dodero *et al.* at 162.2 and 164.1 eV [49]). Moreover, after the chelation of Ag ions, the B.E.s of nitrogen and sulfur electrons in SH-PANF are decreased, which indicates that more electronegative N and S atoms were detected. This suggests that the reaction between –NH, –SH and S-S with Ag ions increases the electron density around N and S centres [32,48,49]. Deconvolution of the high-resolution XPS profile of Ag3$d$ (figure 7$g$) reveals two new strong peaks at 367.82 and 368.42 eV for Ag-SH-PANF, indicating that Ag was successfully chelated. These are attributed to silver ions bound to amino, sulfhydryl and disulfide groups, respectively [50,51]. Consequently, we could confirm that the adsorption of Ag was achieved by its chelation with the amino, sulfhydryl and disulfide groups of SH-PANF. Two additional peaks at higher B.E.s, 373.82 and 374.42 eV, were also detected and were assigned to Ag as well. The spacings between the four peaks for Ag are in line with the predicted energy differences according to the spin-orbit splitting of Ag3$d$, in which the doublet of Ag3$d$ can be well fitted into two peaks for Ag3$d_{5/2}$ and Ag3$d_{3/2}$, with a separation due to spin-orbit splitting of 6.0 eV [51].

## 3.6. Antibacterial activities

In this study, the antibacterial activities of the different fibres were tested by the improved shake flask method; the results are shown in figure 8. PANF and SH-PANF demonstrate no antibacterial performance against the tested pathogens, as set by the National Standard of China (GB/T 20944.3-2008), which states that fibres have antimicrobial activity when the antimicrobial ratio is greater than or equal to 70% against *E. coli* and *S. aureus* and greater than or equal to 60% against *C. albicans* [52]. Hence, we focused only on Ag-SH-PANF, and we compared its antibacterial activities for different Ag contents and exposure times. For *E. coli*, none of the Ag-SH-PANF samples showed antibacterial activity after 30 min contact. Within 1 h, only Ag-SH-PANF 10 (containing 17.46% Ag) proved effective, with an antibacterial ratio of 85.99%. Upon extending the contact time to 2 h, neither Ag-SH-PANF 1 (containing 2.40% Ag) nor Ag-SH-PANF 2 (4.68% Ag) showed any bactericidal properties, but the antibacterial ratios of the Ag-SH-PANF 3–10 fibres (with Ag contents as reported in table 1) were above 93.69%. Ag-SH-PANF 2 achieved optimal antibacterial activity only after 4 h, when its antibacterial ratio reached 85.78%. Ag-SH-PANF 1 proved effective only after 8 h, with a corresponding antibacterial ratio of 97.23%. When the contact time with *E. coli* was extended to 24 h, all fibres showed excellent antibacterial activities and antibacterial ratios up to 99.9999%. The trends in antibacterial activity against *S. aureus* and *C. albicans* followed that against *E. coli*. With a short duration (30 min), none of the fibres showed activity. Within 1 h, the antibacterial ratios of Ag-SH-PANF 10 against *S. aureus* and *C. albicans* were 89.39% and 80.54%, respectively. Ag-SH-PANF 1 and 2 were not effective against these two pathogens after 2 h, whereas Ag-SH-PANF 3 (8.60% Ag) was inactive against *C. albicans* (fungi are usually more resistant than bacteria). In addition, after 2 h, Ag-SH-PANF 4 (10.93% Ag) produced an antibacterial ratio of only 66.76% against *C. albicans*. After 4 h, Ag-SH-PANF 2 was active towards *S. aureus* but inactive towards *C. albicans*; however, Ag-SH-PANF 1 was not active against either of these two pathogens. Eventually, after 6 h, all the fibres showed antibacterial activities against *S. aureus* and *C. albicans*; the longer the time, the higher

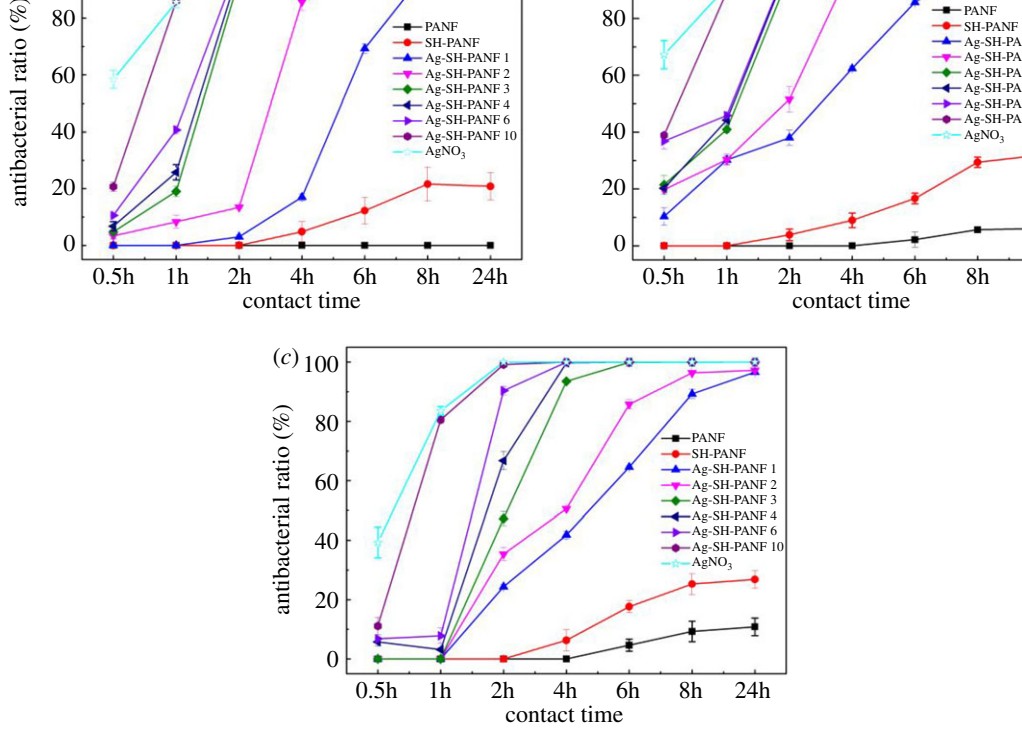

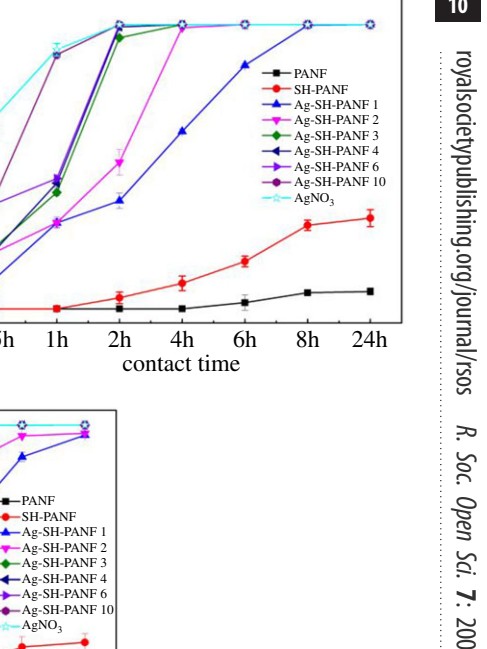

**Figure 8.** Antibacterial activities of Ag-SH-PANF against (a) *E. coli*, (b) *S. aureus* and (c) *C. albicans*.

**Table 2.** Washability of Ag-SH-PANF.

| washing times | | 1 | 5 | 10 | 20 | 50 | 100 |
|---|---|---|---|---|---|---|---|
| antibacterial efficiency | Ag-SH-PANF 1 | 99.9989 | 99.9937 | 99.9932 | 99.9909 | 99.9762 | 98.5926 |
| (%) against *E. coli* | Ag-SH-PANF 10 | 100 | 99.9999 | 99.9966 | 99.9956 | 99.9938 | 99.1777 |

the antibacterial activity. In summary, the antibacterial activities of these fibres were directly proportional to their silver contents and the exposure time to the pathogens. For comparison, the antibacterial properties of AgNO$_3$ were also investigated; the results are shown in figure 8. Further, the used silver mass of AgNO$_3$ was the same as the content of silver in 0.1 g Ag-SH-PANF 10. The results indicated that AgNO$_3$ also did not exhibit antibacterial performance against the tested pathogens after 30 min, but its antibacterial ratios were much higher than Ag-SH-PANF 10. Within 1 h, the antibacterial ratios of AgNO$_3$ against *E. coli*, *S. aureus* and *C. albicans* were 85.65%, 91.18% and 83.58%, respectively, and the difference between Ag-SH-PANF 10 and AgNO$_3$ almost disappeared. After 2 h, the antibacterial ratio of AgNO$_3$ reached up to 100% against all the tested pathogens. Therefore, the antibacterial trends of Ag-SH-PANF were consistent with AgNO$_3$. Moreover, the antibacterial activities of the fibres were the same as AgNO$_3$ against the tested pathogens and followed the order: *S. aureus* > *E. coli* > *C. albicans*.

## 3.7. Durability to washing

The Ag-SH-PANF fibres were washed multiple times in a 0.1% neutral hand detergent for 20 min, then rinsed with distilled water and dried at 40°C overnight. A fibre sample (0.1 g) after this treatment was subjected to the antibacterial tests. The results shown in table 2 reveal that the antibacterial ratios of Ag-SH-PANF exposed to *E. coli* for 24 h only decrease slightly with the increase in the number of wash cycles. However, the antibacterial activity is retained even after 100 washes. The antibacterial ratios of Ag-SH-PANF 1 and 10 remain above 98%, indicating that these fibres are endowed with

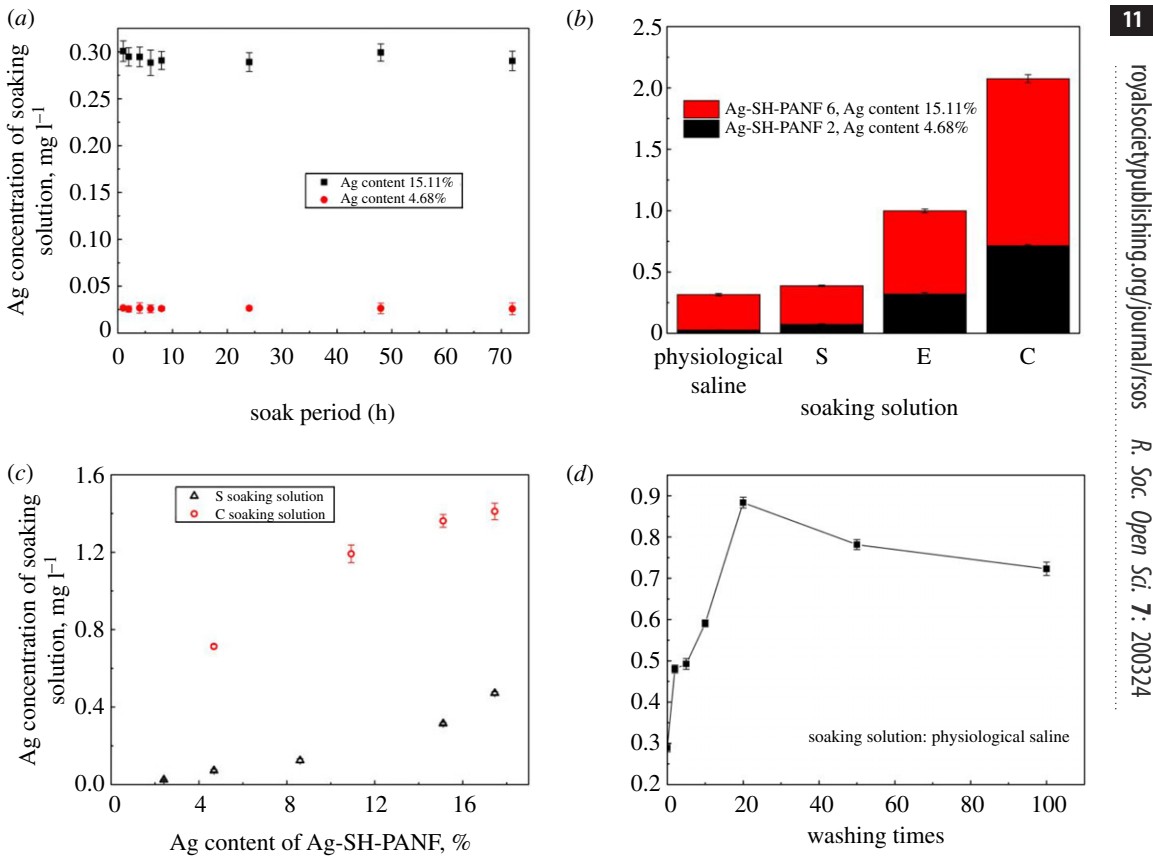

**Figure 9.** Effects of soak period (*a*), soaking solution (*b*), Ag content (*c*) and washing times (*d*) on the release of silver from Ag-SH-PANF. (S, *S. Aureus*; E, *E. coli*; C, *C. albicans*.)

remarkable washing resistance. Thus, repeated washing cycles did not affect antibacterial activity, proving that Ag-SH-PANF have potential to be used as durable antibacterial fibre-based materials.

## 3.8. Antibacterial mechanism of polyacrylonitrile fibres with chelated Ag ions

To study the mechanism underlying the antibacterial activity of Ag-SH-PANF, experiments on the effects of different soaking solutions and soaking periods on Ag release were carried out; figure 9 clearly displays the results. In figure 9*a,c*, the release of Ag is seen to increase with increasing Ag content of the fibre. This trend is not associated with the type of the soaking solution, nor does the soaking period have any evident effect on Ag release. The concentration of Ag ions in the soaking solutions reaches equilibrium after 1 h. According to figure 9*b*, different soaking solutions had distinctive effects on the release of Ag, and the concentration of released Ag found in different soaking solutions was consistent with the antibacterial activities of the fibres (*S. aureus* > *E. coli* > *C. albicans*). This can be correlated with the observation that the Ag concentration of a fibre has a significant effect on its antibacterial activity. In figure 9*d*, we observe that the number of washings notably affects Ag release, especially after 20 cycles, which may be an effect of the detergent used. However, after 20 cycles, the Ag concentration in the soaking solution decreases only slightly and becomes stable until 100 washing cycles. Subsequently, the Ag concentration in the soaking solution maintains a respectable and somewhat constant value, which also ensures superior antibacterial activity.

These results show that the antibacterial activities of Ag-SH-PANF are closely related to the release of Ag in the solution, which is consistent with the phenomenon reported by Sun *et al.* [11]. The results also show that the antibacterial mechanism of Ag-SH-PANF is similar to that of Ag ions in solution, which has been reported in many studies [8,10,11]. In these studies, the antibacterial mechanism can be explained by a complex process that occurs when Ag ions encounter pathogens. When Ag ions combine with electron-donating receptors (especially those bearing imidazole, sulfhydryl, carbonyl, amino, disulfide and phosphate moieties) on cytomembranes and nucleic acids, intracellular absorption and membrane-related enzyme inactivation can ensue. The membrane-related enzyme

inactivation leads to denaturation of the bacterial cell's envelope, which is ultimately lethal to the cell [10,26,53]. Moreover, Davies & Etris [54] reported that microorganisms are not directly attacked by Ag ions. Ag ions act as catalysts and are not consumed during the antibacterial process, but they can accelerate the antibacterial process if present in a certain concentration range. This principle explains why even Ag-SH-PANF with low Ag contents display very good antibacterial properties after being in contact with bacteria for 24 h, whereas, fibres with high Ag contents achieve good antibacterial performance more quickly.

# 4. Conclusion

In this study, we successfully prepared a series of new antibacterial fibres, Ag-SH-PANF, with different Ag contents by a grafting and chelation sequence. The presence of Ag ions on the surface of Ag-SH-PANF was confirmed by XPS and FT-IR spectroscopy, which showed that chelation takes place between Ag ions and amino, sulfhydryl and disulfide groups. Moreover, XRD analysis confirmed that no silver crystals were produced on Ag-SH-PANF. Antibacterial testing showed that Ag-SH-PANF are excellent antibacterial agents against *S. aureus*, but less effective against *E. coli*, which is ascribed to structural differences between the two bacteria. Furthermore, we observed that the antibacterial activity of the fibres increased with the contact time between bacteria and fibres, as well as with the initial Ag content of the fibres. In addition, the results of washing durability experiments indicated the good washing durability of Ag-SH-PANF, whose antibacterial efficiency against *E. coli* remained above 98% after washing the fibres 100 times, even though the Ag content of the fibre was only 2.40%. Finally, silver release experiments elucidated the antibacterial mechanism of Ag-SH-PANF was the same as silver ions, in which silver ions are the main antibacterial factors, acting as catalysts and therefore are not consumed in the antibacterial process. Nonetheless, a relatively high concentration of silver ions can accelerate bacterial cell death within a certain concentration range.

Data accessibility. Our data are deposited at Dryad Digital Repository: https://doi.org/10.5061/dryad.3tx95x6c6 [55].

Authors' contributions. L.C. and W.D. contributed equally to this work; they participated in all the experiments, did most of the data analysis and wrote the manuscript; A.C. and L.J.L. prepared the fibres and completed the characterization of the samples; S.H., J.T., G.P. and Y.D. completed the antibacterial experiments; L.Z. and D.L. conceived the research and provided the necessary experimental equipment. L.Z. assisted the research, especially in the preparation of fibres. All authors gave final approval for publication.

Competing interests. We declare we have no competing interests.

Funding. This work was supported by the Natural Science Foundation of Hunan Province (2018JJ3584) and Henan Environmental Functional Materials Outstanding Foreign Scientist Studio Funding (GZS2018002).

Acknowledgements. We are particularly grateful to the Key Laboratory of Microbial Engineering, Henan Academy of Sciences Institute of Biology, for supporting our antibacterial tests.

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
