## [Reviewer comments · Royal Society Open Science]

Review History

RSOS-200324.R0 (Original submission)

Review form: Reviewer 1

Is the manuscript scientifically sound in its present form?

No

Are the interpretations and conclusions justified by the results?

No

Is the language acceptable?

Yes

Do you have any ethical concerns with this paper?

No

Have you any concerns about statistical analyses in this paper?

No

Recommendation?

Major revision is needed (please make suggestions in comments)

Comments to the Author(s)

This manuscript reported a series of novel polyacrylonitrile-based fibres with chelated Ag ions and their antibacterial activities. However, the synthesized methods are not brand new, besides the biological activity research is not deep enough. Thus, I do not think it is suitable to publish on Royal Society Open Science in present status.

The authors should consider correction of the following suggestions:

1. In the summary part and part 3.8, the author mentioned the antibacterial mechanism, however, according to the report the fibres' antibacterial activity entirely depends on the Ag ions. There is no real mechanism research part in this paper, just a relationship of antibacterial activities and the concentration of Ag.
2. In the part of 3.2, the absorption peak from 3150 to 3700 cm^{-1} was mainly due to the water in the sample. Why not dry the sample? In that case, it is not hard to figure out the peak of N-H.
3. The IR spectra of Ag-SH-PANF was almost the same as that of SH-PANF, how can the author confirm the Ag ion is chelated with the ligand? The slight red-shift of 3 cm^{-1} could not be a proof.
4. Where is the peak of S-S bond in IR spectra?
5. In the antibacterial part, the positive group of Ag ion should be compared.
6. Also, in part 2.5 how can the author confirm the release Ag ion is came from the chelated Ag ion, rather than the scatter ones in the fibres?
7. By the way, do the fibres produce antibacterial activity only in solution? In that case, they can release Ag ion. Thus, how can we use them as a new medical material? Could the release Ag ion in dry condition?

Review form: Reviewer 2

Is the manuscript scientifically sound in its present form?

Yes

Are the interpretations and conclusions justified by the results?

Yes

Is the language acceptable?

Yes

Do you have any ethical concerns with this paper?

Yes

Have you any concerns about statistical analyses in this paper?

No

Recommendation?

Accept with minor revision (please list in comments)

Comments to the Author(s)

The paper reports on fiber-silver complex for antibacterial application. The paper is interesting. Some points need be strengthened. There are also some concerns to be addressed. All these are stated as follows:

Why was this material selected for antibacterial studies? There have been many others, this needs be defined and some recent typical papers should be cited: Bioactive compounds: antioxidant, antibacterial and antiproliferative activities in chloranthus henryi, *Sci. Adv. Mater.*, 2020, 12, 144–151; Durably antibacterial and bacterially anti-adhesive cotton fabrics coated by cationic fluorinated polymers, *ACS Appl. Mater. Interfaces*, 2018, 10, 6124–6136; A new anti-biofilm

strategy of enabling arbitrary surfaces of materials and devices with robust bacterial anti-adhesion via spraying modified microspheres method, *J. Mater. Chem. A*, 2019, 7, 26039; Antifouling and antibacterial behaviors of capsaicin-based pH responsive smart coatings in marine environments, *Mater. Sci. Eng. C*, 2020, 108, 110361;

The application of the PAN-based fibers should be introduced: Chemical modification of carbon fiber with diethylenetriaminepentaacetic acid/halloysite nanotube as a multifunctional interfacial reinforcement for silicone resin composites. *Polym. Adv. Technol.*, 2020, 31, 527; Modification of renewable cardanol onto carbon fiber for the improved interfacial properties of advanced polymer composites. *Polymers*, 2020, 12, 45; Improved Thermal Stabilities, Ablation and Mechanical Properties for Carbon Fibers/Phenolic Resins Laminated Composites Modified by Silicon-containing Polyborazine, *Engineered Science*, 2018, 2, 57-66; Effect of A Vinyl Ester-Carbon Nanotubes Sizing Agent on Interfacial Properties of Carbon Fibers Reinforced Unsaturated Polyester Composites, *ES Materials & Manufacturing*, 2019, 6, 38-48; . Effect of MoO₃/carbon nanotubes on friction and wear performance of glass fabric-reinforced epoxy composites under dry sliding, *Appl. Surf. Sci.*, 2020, 506, 144946; Reinforcing carbon fiber epoxy composites with triazine derivatives functionalized graphene oxide modified sizing agent, *Compos. Part B*, 2019, 176, 107078.

For broad impact, the applications of silver nanoparticles on other substrate need be introduced including catalysts: Fast 4-nitrophenol Reduction Using Gelatin Hydrogel Containing Silver Nanoparticles, *Engineered Science*, 2019, 8, 19-24; Facile One-pot Synthesis of Silver Nanoparticles Supported on α -Zirconium Phosphate Single-Layer Nanosheets, *ES Materials & Manufacturing*, 2019, 5, 24-28; Reduced graphene oxide heterostructured silver nanoparticles significantly enhanced thermal conductivities in hot-pressed electrospun polyimide nanocomposites, *ACS Appl. Mater. Interfaces*, 2019, 11, 25465; Interfacially reinforced carbon fiber silicone resin via constructing functional nano-structural silver, *Compos. Sci. Technol.*, 2019, 181, 107689.

The TGA part needs be discussed with more details and the following papers will be helpful: Investigation of Thermostability of Modified Graphene Oxide/Methylsilicone Resin Nanocomposites, *Engineered Science*, 2019, 5, 73-78; Micro-crack behavior of carbon fiber reinforced Fe₃O₄/graphene oxide modified epoxy composites for cryogenic application. *Compos. Part A*, 2018, 108, 12-22.

The conclusion needs be more focused and should be revised.

The language needs be polished for a precise presentation.

Review form: Reviewer 3

Is the manuscript scientifically sound in its present form?

No

Are the interpretations and conclusions justified by the results?

Yes

Is the language acceptable?

No

Do you have any ethical concerns with this paper?

No

Have you any concerns about statistical analyses in this paper?

No

Recommendation?

Major revision is needed (please make suggestions in comments)

Comments to the Author(s)

Authors have reported the synthesis of polyacrylonitrile-based fibres with chelated with Ag⁺ ions (Ag-SH-PANF) and its antibacterial activities has been studied against pathogenic bacteria. Although results are interesting, several areas need major modifications before this manuscript can be considered for publication. See the comments below:

It is stated in the experimental section that the release of Ag amount from the Ag-SH-PANF was tested by treating with soaking solutions. What is the soaking solution? How long the treatment was carried out? How the authors are sure that all Ag ions are released from the polymer after this treatment?

Introduction is not written properly and too brief. Authors have functionalized polyacrylonitrile fibres with surface S-S sites for the chelation of Ag(I), largely due to soft acid-soft base interactions. Ag⁺/Hg²⁺ favourably bind with solid matrixes bearing soft sulphur containing ligand sites. This should be discussed in the background. See and include such interactions described by different research groups: ACS Sustainable Chem. Eng. 2019, 7, 7353-7361; Chem. Eng. J. 2018, 332, 387-397.

Resolution of the electron microscopic images are quite low. Authors should provide more high resolution images of Ag-SH-PANF, which can provide better understanding on the surface nanostructure.

Decision letter (RSOS-200324.R0)

Dear Dr Chang:

Title: Preparation of Polyacrylonitrile-based Fibres with Chelated Ag ions for Antibacterial Applications
Manuscript ID: RSOS-200324

The editor assigned to your manuscript has now received comments from reviewers. We would like you to revise your paper in accordance with the referee and Subject Editor suggestions which can be found below (not including confidential reports to the Editor). Please note this decision does not guarantee eventual acceptance.

Please submit your revised paper before 28-May-2020. Please note that the revision deadline will expire at 00.00am on this date. If we do not hear from you within this time then it will be assumed that the paper has been withdrawn. In exceptional circumstances, extensions may be possible if agreed with the Editorial Office in advance. We do not allow multiple rounds of revision so we urge you to make every effort to fully address all of the comments at this stage. If deemed necessary by the Editors, your manuscript will be sent back to one or more of the original reviewers for assessment. If the original reviewers are not available we may invite new reviewers.

To revise your manuscript, log into <http://mc.manuscriptcentral.com/rsos> and enter your Author Centre, where you will find your manuscript title listed under "Manuscripts with

Decisions." Under "Actions," click on "Create a Revision." Your manuscript number has been appended to denote a revision. Revise your manuscript and upload a new version through your Author Centre.

RSC Associate Editor:
Comments to the Author:
(There are no comments.)

RSC Subject Editor:
Comments to the Author:
(There are no comments.)

Reviewers' Comments to Author:
Reviewer: 1

Comments to the Author(s)

This manuscript reported a series of novel polyacrylonitrile-based fibres with chelated Ag ions and their antibacterial activities. However, the synthesized methods are not brand new, besides the biological activity research is not deep enough. Thus, I do not think it is suitable to publish on Royal Society Open Science in present status.

The authors should consider correction of the following suggestions:

1. In the summary part and part 3.8, the author mentioned the antibacterial mechanism, however, according to the report the fibres' antibacterial activity entirely depends on the Ag ions. There is no real mechanism research part in this paper, just a relationship of antibacterial activities and the concentration of Ag.
2. In the part of 3.2, the absorption peak from 3150 to 3700 cm^{-1} was mainly due to the water in the sample. Why not dry the sample? In that case, it is not hard to figure out the peak of N-H.
3. The IR spectra of Ag-SH-PANF was almost the same as that of SH-PANF, how can the author confirm the Ag ion is chelated with the ligand? The slight red-shift of 3 cm^{-1} could not be a proof.

4. Where is the peak of S-S bond in IR spectra?
5. In the antibacterial part, the positive group of Ag ion should be compared.
6. Also, in part 2.5 how can the author confirm the release Ag ion is came from the chelated Ag ion, rather than the scatter ones in the fibres?
7. By the way, do the fibres produce antibacterial activity only in solution? In that case, they can release Ag ion. Thus, how can we use them as a new medical material? Could the release Ag ion in dry condition?

Reviewer: 2

Comments to the Author(s)

The paper reports on fiber-silver complex for antibacterial application. The paper is interesting. Some points need be strengthened. There are also some concerns to be addressed. All these are stated as follows:

Why was this material selected for antibacterial studies? There have been many others, this needs be defined and some recent typical papers should be cited: Bioactive compounds: antioxidant, antibacterial and antiproliferative activities in chloranthus henryi, *Sci. Adv. Mater.*, 2020, 12, 144–151; Durably antibacterial and bacterially anti-adhesive cotton fabrics coated by cationic fluorinated polymers, *ACS Appl. Mater. Interfaces*, 2018, 10, 6124–6136; A new anti-biofilm strategy of enabling arbitrary surfaces of materials and devices with robust bacterial anti-adhesion via spraying modified microspheres method, *J. Mater. Chem. A*, 2019, 7, 26039; Antifouling and antibacterial behaviors of capsaicin-based pH responsive smart coatings in marine environments, *Mater. Sci. Eng. C*, 2020, 108, 110361;

The application of the PAN-based fibers should be introduced: Chemical modification of carbon fiber with diethylenetriaminepentaacetic acid/halloysite nanotube as a multifunctional interfacial reinforcement for silicone resin composites. *Polym. Adv. Technol.*, 2020, 31, 527; Modification of renewable cardanol onto carbon fiber for the improved interfacial properties of advanced polymer composites. *Polymers*, 2020, 12, 45; Improved Thermal Stabilities, Ablation and Mechanical Properties for Carbon Fibers/Phenolic Resins Laminated Composites Modified by Silicon-containing Polyborazine, *Engineered Science*, 2018, 2, 57–66; Effect of A Vinyl Ester-Carbon Nanotubes Sizing Agent on Interfacial Properties of Carbon Fibers Reinforced Unsaturated Polyester Composites, *ES Materials & Manufacturing*, 2019, 6, 38-48; . Effect of MoO₃/carbon nanotubes on friction and wear performance of glass fabric-reinforced epoxy composites under dry sliding, *Appl. Surf. Sci.*, 2020, 506, 144946; Reinforcing carbon fiber epoxy composites with triazine derivatives functionalized graphene oxide modified sizing agent, *Compos. Part B*, 2019, 176, 107078.

For broad impact, the applications of silver nanoparticles on other substrate need be introduced including catalysts: Fast 4-nitrophenol Reduction Using Gelatin Hydrogel Containing Silver Nanoparticles, *Engineered Science*, 2019, 8, 19-24; Facile One-pot Synthesis of Silver Nanoparticles Supported on α -Zirconium Phosphate Single-Layer Nanosheets, *ES Materials & Manufacturing*, 2019, 5, 24-28; Reduced graphene oxide heterostructured silver nanoparticles significantly enhanced thermal conductivities in hot-pressed electrospun polyimide nanocomposites, *ACS Appl. Mater. Interfaces*, 2019, 11, 25465; Interfacially reinforced carbon fiber silicone resin via constructing functional nano-structural silver, *Compos. Sci. Technol.*, 2019, 181, 107689.

The TGA part needs be discussed with more details and the following papers will be helpful: Investigation of Thermostability of Modified Graphene Oxide/Methylsilicone Resin Nanocomposites, *Engineered Science*, 2019, 5, 73-78; Micro-crack behavior of carbon fiber reinforced Fe₃O₄/graphene oxide modified epoxy composites for cryogenic application. *Compos. Part A*, 2018, 108, 12-22.

The conclusion needs be more focused and should be revised.

The language needs be polished for a precise presentation.

Reviewer: 3

Comments to the Author(s)

Authors have reported the synthesis of polyacrylonitrile-based fibres with chelated with Ag⁺ ions (Ag-SH-PANF) and its antibacterial activities has been studied against pathogenic bacteria. Although results are interesting, several areas need major modifications before this manuscript can be considered for publication. See the comments below:

It is stated in the experimental section that the release of Ag amount from the Ag-SH-PANF was tested by treating with soaking solutions. What is the soaking solution? How long the treatment was carried out? How the authors are sure that all Ag ions are released from the polymer after this treatment?

Introduction is not written properly and too brief. Authors have functionalized polyacrylonitrile fibres with surface S-S sites for the chelation of Ag(I), largely due to soft acid-soft base interactions. Ag⁺/Hg²⁺ favourably bind with solid matrixes bearing soft sulphur containing ligand sites. This should be discussed in the background. See and include such interactions described by different research groups: ACS Sustainable Chem. Eng. 2019, 7, 7353-7361; Chem. Eng. J. 2018, 332, 387-397.

Resolution of the electron microscopic images are quite low. Authors should provide more high resolution images of Ag-SH-PANF, which can provide better understanding on the surface nanostructure.

Author's Response to Decision Letter for (RSOS-200324.R0)

See Appendix A.

Decision letter (RSOS-200324.R1)

Dear Dr Chang:

Title: Preparation of Polyacrylonitrile-based Fibres with Chelated Ag ions for Antibacterial Applications
Manuscript ID: RSOS-200324.R1

It is a pleasure to accept your manuscript in its current form for publication in Royal Society Open Science. The chemistry content of Royal Society Open Science is published in collaboration with the Royal Society of Chemistry.

RSC Associate Editor
Comments to the Author:
(There are no comments.)

Reviewer(s)' Comments to Author:

Appendix A

Dear editor and reviewers:

Thanks for your valuable advices. In view of the reviewers' comments, we have made the necessary changes to the manuscript, which have been marked in red. In addition, the data deposited in Dryad has also been simultaneously updated. Besides, a point-by-point response to the reviewers' comments is provided below.

Response to reviewers' comments

Manuscript ID: RSOS-200324

TITLE: Preparation of Polyacrylonitrile-based Fibres with Chelated Ag ions for Antibacterial Applications

Referee: 1

Comments to the Author(s)

This manuscript reported a series of novel polyacrylonitrile-based fibres with chelated Ag ions and their antibacterial activities. However, the synthesized methods are not brand new, besides the biological activity research is not deep enough. Thus, I do not think it is suitable to publish on Royal Society Open Science in present status.

The authors should consider correction of the following suggestions:

1. In the summary part and part 3.8, the author mentioned the antibacterial mechanism, however, according to the report the fibres' antibacterial activity entirely depends on the Ag ions. There is no real mechanism research part in this paper, just a relationship of antibacterial activities and the concentration of Ag.
2. In the part of 3.2, the absorption peak from 3150 to 3700 cm^{-1} was mainly due to the water in the sample. Why not dry the sample? In that case, it is not hard to figure out the peak of N-H.
3. The IR spectra of Ag-SH-PANF was almost the same as that of SH-PANF, how can the author confirm the Ag ion is chelated with the ligand? The slight red-shift of 3 cm^{-1} could not be a proof.
4. Where is the peak of S-S bond in IR spectra?
5. In the antibacterial part, the positive group of Ag ion should be compared.
6. Also, in part 2.5 how can the author confirm the release Ag ion is came from the chelated Ag ion, rather than the scatter ones in the fibres?
7. By the way, do the fibres produce antibacterial activity only in solution? In that case, they can release Ag ion. Thus, how can we use them as a new medical material? Could the release Ag ion in dry condition?

REPLY: Thank you for your valuable comments and suggestions.

1. In this study, a series of polyacrylonitrile-based fibres with chelated silver ions (Ag-SH-PANF) were prepared, and their excellent antibacterial performance was confirmed. To identify whether the fibres act as an in situ antibacterial or agent-releasing antibacterial material, the Ag ion release experiment was designed, and the relationship between the amount of silver ions released and the antibacterial performance was determined. Thus, it was shown that the antibacterial performance of the material (i.e. Ag-SH-PANF) was achieved through the antibacterial agent. Besides, the antibacterial mechanism of Ag-SH-PANF is consistent with that of silver ions. Moreover, the antibacterial mechanism of Ag ions has been investigated by many researchers (please see, references 18 and 34-37 in the manuscript).
2. We dried the fibres for 4 h at 60°C before FT-IR test. PANF used in this study are commercial fibres, in which the addition of the second and third monomers improves their moisture absorption performance. However, the chemical modification of PANF introduces a large number of amino, sulfhydryl and disulfide groups on their surface. Further, the introduction of amino, sulfhydryl and disulfide groups further on the surface of PANF further enhances their moisture absorption performance. In addition, the as-obtained FT-IR spectra were collected using the attenuated total reflection (ATR) method, and the absorption peaks in the ATR-FT-IR spectra were not as strong as those in the FT-IR spectra collected using the KBr pellet method. Therefore, we obtained new FT-IR spectra for the three fibres using the KBr pellet method, and the as-obtained FT-IR spectra were replaced. Further, the corresponding results have also been modified accordingly in section 3.2, and are marked in red. Unfortunately, although we tried our best to avoid the influence of water, it was still observed in the new spectra.

MODIFICATION:

The new FT-IR spectra for the three fibres using the KBr pellet method were showed in Fig. 3. The related sentences in section 2.3 were modified as "FT-IR spectra were collected with a Nicolet IR200 spectrometer (Thermo Electron Scientific Instruments, USA) in the range of 4000–400 cm^{-1} ; samples were prepared by the KBr pellet method".

Fig. 3 FT-IR spectra of PANF, SH-PANF, and Ag-SH-PANF.

- The new spectra obtained by KBr pellet method showed distinct changes before and after Ag chelation. For Ag-SH-PANF, the spectrum showed a narrower absorption peak at 3150–3700 cm^{-1} and a stronger and sharper peak at 1384 cm^{-1} that can be ascribed to the red-shift of the peak at 1389 cm^{-1} (C–N) in the spectrum of SH-PANF. In addition, a new weak absorption peak appears at 2424 cm^{-1} in the spectrum of Ag-SH-PANF, which belongs to the sulfhydryl group. However, this peak did not appear in the spectrum of SH-PANF. All these phenomena can be attributed to the formation of complexes between amino and sulfhydryl groups with silver.
- The new FT-IR spectra obtained using the KBr pellet method showed that a new weak absorption peak appeared at 520 cm^{-1} in the spectrum of SH-PANF, which belonged to the disulfide group for SH-PANF. However, this peak disappeared in the spectrum of Ag-SH-PANF. In addition, the XPS characterization also confirmed the existence of S-S (discussed in section 3.5). “Deconvolution of the high-resolution XPS profile of $S2p$ (Fig.7(d) and (f)) reveals the presence of multiple peaks, assignable to the sulfur atoms in sulfhydryl, disulfide, and sulfo groups [47, 48] at 163.64 and 165.08 eV, 164.40 eV, and 168.83 eV, respectively.”
- We have supplemented the antibacterial data of silver ions, and made the corresponding modification in section 3.6.

MODIFICATION:

The sentences and Figures related to the experimental results have been added in section 3.6 and Fig. 8, respectively. The sentences are as follows: “For comparison, the antibacterial properties of AgNO_3 were also investigated; the results are shown in Fig. 8. Further, the used silver mass of AgNO_3 was the same as the content of silver in 0.1 g Ag-SH-PANF 10. The results indicated that AgNO_3 also did not exhibit antibacterial performance against the tested pathogens after 30 min, but its antibacterial ratios were much higher than Ag-SH-PANF 10. Within 1 h, the antibacterial ratios of AgNO_3 against *E. coli*, *S. aureus* and *C. albicans* were 85.65%, 91.18% and 83.58%, respectively, and the difference between Ag-SH-PANF 10 and AgNO_3 almost disappeared. After 2 h, the antibacterial ratio of AgNO_3 reached up to 100% against all the tested pathogens. Therefore, the antibacterial trends of Ag-SH-PANF were consistent with AgNO_3 . Moreover, the antibacterial activities of the fibres were the same as AgNO_3 against the tested pathogens and followed the order: *S. aureus* > *E. coli* > *C. albicans*”.

(1) Marked as 0 when the antibacterial ratio is negative. (2) The used silver mass in AgNO₃ is the same as the content of silver in 0.1g Ag-SH-PANF 10

Fig. 8 Antibacterial activities of Ag-SH-PANF against (a) *E. coli*, (b) *S. aureus*, and (c) *C. albicans*.

- The XRD characterization confirmed that no silver crystals were produced on Ag-SH-PANF. Moreover, the prepared Ag-SH-PANF were washed with deionised water until no more Ag ions could be detected in washwater. Therefore, there were no scattered silver ions in the fibres.
- The antibacterial activity test was performed according to Chinese national standard "Textiles - Evaluation for anti bacterial activity - Part 3: Shake flask method" (GB/T 20944.3-2008). It simulates human clothing and living conditions. In particular, it focuses on textiles such as bedding, underwear, and socks, which are often in contact with human skin and absorb sweat. Therefore, the fibres won't release silver ions in absolutely dry condition, but release in a humid condition.

Reviewer: 2

Comments to the Author(s)

The paper reports on fiber-silver complex for antibacterial application. The paper is interesting. Some points need be strengthened. There are also some concerns to be addressed. All these are stated as follows:

Why was this material selected for antibacterial studies? There have been many others, this needs be defined and some recent typical papers should be cited: Bioactive compounds: antioxidant, antibacterial and antiproliferative activities in chloranthus henryi, *Sci. Adv. Mater.*, 2020, 12, 144–151; Durably antibacterial and bacterially anti-adhesive cotton fabrics coated by cationic fluorinated polymers, *ACS Appl. Mater. Interfaces*, 2018, 10, 6124–6136; A new anti-biofilm strategy of enabling arbitrary surfaces of materials and devices with robust bacterial anti-adhesion via spraying modified microspheres method, *J. Mater. Chem. A*, 2019, 7, 26039; Antifouling and antibacterial behaviors of capsaicin-based pH responsive smart coatings in marine environments, *Mater. Sci. Eng. C*, 2020, 108, 110361;

The application of the PAN-based fibers should be introduced: Chemical modification of carbon fiber with diethylenetriaminepentaacetic acid/halloysite nanotube as a multifunctional interfacial reinforcement for silicone resin composites. *Polym. Adv. Technol.*, 2020, 31, 527; Modification of renewable cardanol onto carbon fiber for the improved interfacial properties of advanced polymer composites. *Polymers*, 2020, 12, 45; Improved Thermal Stabilities, Ablation and Mechanical Properties for Carbon Fibers/Phenolic Resins Laminated Composites Modified by Silicon-containing Polyborazine, *Engineered Science*, 2018, 2, 57–66; Effect of A Vinyl Ester-Carbon Nanotubes Sizing Agent on Interfacial Properties of Carbon Fibers Reinforced Unsaturated Polyester Composites, *ES Materials & Manufacturing*, 2019, 6, 38-48; . Effect of MoO₃/carbon nanotubes on friction and wear performance of glass fabric-reinforced epoxy composites under dry sliding, *Appl. Surf. Sci.*, 2020, 506, 144946; Reinforcing carbon fiber epoxy composites with triazine derivatives functionalized graphene oxide modified sizing agent, *Compos. Part B*, 2019, 176, 107078.

For broad impact, the applications of silver nanoparticles on other substrate need be introduced including catalysts: Fast 4-nitrophenol Reduction Using Gelatin Hydrogel Containing Silver Nanoparticles, *Engineered Science*, 2019, 8, 19-24; Facile One-pot Synthesis of Silver Nanoparticles Supported on α -Zirconium Phosphate Single-Layer Nanosheets, *ES Materials & Manufacturing*, 2019, 5, 24-28; Reduced graphene oxide heterostructured silver nanoparticles significantly enhanced thermal conductivities in hot-pressed electrospun polyimide nanocomposites, *ACS Appl. Mater. Interfaces*, 2019, 11, 25465; Interfacially reinforced carbon fiber silicone resin via constructing functional nano-structural silver, *Compos. Sci. Technol.*, 2019, 181, 107689.

The TGA part needs be discussed with more details and the following papers will be helpful: Investigation of Thermostability of Modified Graphene Oxide/Methylsilicone Resin Nanocomposites, *Engineered Science*, 2019, 5, 73-78; Micro-crack behavior of carbon fiber reinforced Fe₃O₄/graphene oxide modified epoxy composites for

cryogenic application. Compos. Part A, 2018, 108, 12-22.
The conclusion needs be more focused and should be revised.
The language needs be polished for a precise presentation.

REPLY: Thank you for your valuable comments and suggestions.

The corresponding changes have been made in section 1, section 3.3, and section 4, and more papers have been cited. These changes are marked in red.

We have polished the language for a precise presentation, and the relevant changes are marked in red.

MODIFICATION:

The sentences we added in section 1 are as follows: " Various antibacterial components are used for preparing antibacterial materials, such as quaternary ammonium groups [5], quaternary phosphonium groups [6, 7], silver nanoparticles [8, 9], silver ions [10, 11], extracts from *Chloranthus henryi* [12], and capsaicin [13]" and "This makes PANF an ideal raw material to prepare functional fibres, such as amine [30], quaternary ammonium [31], quaternary phosphonium [7], sulfhydryl [32], carboxyl [33], carbon [34-37], and amidoxime [38] fibres, by chemical modification".

The section 3.3 after modification is as follows: "The TGA results for PANF, SH-PANF, and Ag-SH-PANF are presented in Fig. 5 (a) and (b). It can be easily seen from Fig. 5(b) that the weight loss rate of PANF is much higher than those of other two fibres. According to Fig. 5(a), in the case of PANF, there are two distinct plateaus in the TGA curve, with the first decomposition temperature near 290 °C. The cyclisation of -CN groups contributes to the first decomposition plateau, and the formation of a cross-linked structure in response to the oxidation and dehydrogenation of the functionalities on the macromolecular chains contributes to the second decomposition plateau [44]. After the desorption of atmospheric gases and the loss of water near 120°C, only one prominent plateau is observed in the curve for SH-PANF, whereas two plateaus appear again for Ag-SH-PANF. The second plateaus in SH-PANF and Ag-SH-PANF correspond to the first plateau of PANF. However, the decomposition temperatures for SH-PANF and Ag-SH-PANF are much lower than that of PANF, which is ascribed to the consumption of most -CN groups and the simultaneous conversion to other functionalities (i.e. -C=N) during the grafting process. The third plateau reflects the behaviour of the grafted cysteamine molecules and the cross-linked, oxidised, and dehydrogenated macromolecular chains. However, after the whole process, Ag-SH-PANF gives a significantly higher char yield of 69.2% than SH-PANF (56.5%) and PANF (49.4%), which may be attributed to the large amounts of adsorbed silver. This finding accords with those of Duan W et al. [32], but are contrary to those stated by Sun C et al. [45], who observed that the char yield of the unmodified fibres was higher than that of heavy-metal-loaded fibres".

The section 4 after modification is as follows: "In this study, we successfully prepared a series of new antibacterial fibres, Ag-SH-PANF, with different Ag contents by a grafting and chelation sequence. The presence of Ag ions on the surface of Ag-SH-PANF was confirmed by XPS and FT-IR spectroscopy, which showed that chelation takes place between Ag ions and amino, sulfhydryl and disulfide groups. Moreover, XRD analysis confirmed that no silver crystals were produced on Ag-SH-PANF. Antibacterial testing showed that Ag-SH-PANF are excellent antibacterial agents against *S. aureus*, but less effective against *E. coli*, which is ascribed to structural differences between the two bacteria. Furthermore, we observed that the antibacterial activity of the fibres increased with the contact time between bacteria and fibres, as well as with the initial Ag content of the fibres. In addition, the results of washing durability experiments indicated the good washing durability of Ag-SH-PANF, whose antibacterial efficiency against *E. coli* remained above 98% after washing the fibres 100 times, even though the Ag content of the fibre was only 2.40%. Finally, silver release experiments elucidated the antibacterial mechanism of Ag-SH-PANF was the same as silver ions, in which silver ions are the main antibacterial factors, acting as catalysts and therefore are not consumed in the antibacterial process. Nonetheless, a relatively high concentration of silver ions can accelerate bacterial cell death within a certain concentration range".

Reviewer: 3

Comments to the Author(s)

Authors have reported the synthesis of polyacrylonitrile-based fibres with chelated with Ag⁺ ions (Ag-SH-PANF) and its antibacterial activities has been studied against pathogenic bacteria. Although results are interesting, several areas need major modifications before this manuscript can be considered for publication. See the comments below:

It is stated in the experimental section that the release of Ag amount from the Ag-SH-PANF was tested by treating with soaking solutions. What is the soaking solution? How long the treatment was carried out? How the authors are sure that all Ag ions are released from the polymer after this treatment?

Introduction is not written properly and too brief. Authors have functionalized polyacrylonitrile fibres with surface S-S sites for the chelation of Ag(I), largely due to soft acid-soft base interactions. Ag⁺/Hg²⁺ favourably bind with solid matrixes bearing soft sulphur containing ligand sites. This should be discussed in the background. See and include such interactions described by different research groups: ACS Sustainable Chem. Eng. 2019, 7, 7353-7361; Chem. Eng. J. 2018, 332, 387-397.

Resolution of the electron microscopic images are quite low. Authors should provide more high resolution images of Ag-SH-PANF, which can provide better understanding on the surface nanostructure.

REPLY: Thank you for your valuable comments and suggestions.

1. We apologize for missing the relevant details in the process of repeated modification. However, we have added these details to the note of Fig. 9. In addition, we have added the detailed description of soaking solution and soaking period in Section 2.5, and it has been marked in red.

MODIFICATION:

The relevant details added in the note of Fig. 9 are as follows: "S: *S. aureus* solution, E: *E. coli* solution, C: *C. albicans* solution". In addition, the detailed description of soaking solution and soaking period added in Section 2.5 is as follows: "Except for the experiments to determine the effects of different soaking periods on the release of Ag ions, the soaking period was set to be 24 h for all other experiments concerning the effects of different soaking solutions on the release of Ag ions. Physiological saline composed of 0.9% NaCl solution and *S. aureus*, *E. coli*, and *C. albicans* solutions were the prepared corresponding bacterial suspensions (see Section 2.4) were used as the soaking solutions".

2. The FT-IR and XPS characterization confirmed the formation of complexes between amino, sulfhydryl and disulfide groups with silver. Moreover, XRD characterization confirmed that no silver crystals were produced on Ag-SH-PANF. Besides, the prepared Ag-SH-PANF material was washed with deionised water until no more Ag ions could be detected in washwater.

Therefore, there were no scattered silver ions in the fibres. We are sure that the Ag ions released from the polymer are chelated ones. However, the amount of released silver ions is very small to ensure its durable antibacterial performance.

3. Thanks for your valuable suggestion. Our introduction was really quite brief. Besides, we had not mentioned the reason for selecting the sulfur-containing functional groups to chelate silver ions. Moreover, we have revised and supplemented the introduction in section 1, and the relevant changes are marked in red.

MODIFICATION:

The sentences we have supplemented in section 1 are as follows: "According to the hard and soft acid-base theory, silver ions belong to the category of soft acids, which easily coordinate with soft bases. Besides, many studies have reported that a sulfur-containing group as a typical soft base can efficiently coordinate with silver, mercury, and other soft acids [32, 39, 40]. Therefore, chemical modification is a good method to obtain stable silver containing functional materials through the coordination reaction between sulfur-containing functional groups and silver".

4. We replaced the SEM images with high resolution images in Fig. 6. In addition, three images were provided for every fibre, and their magnifications were 2000x, 20000x and 50000x, respectively.

MODIFICATION:

The high resolution SEM images we placed in Fig. 6 are as follows:

1: 2000x; 2: 20000x; 3: 50000x

Fig. 6 SEM images of PANF (A), SH-PANF (B), and Ag-SH-PANF (C).